# Modelling maternal and perinatal risk factors to predict poorly controlled childhood asthma

**Samuel Schäfer**[1,2], **Kevin Wang**[3], **Felicia Sundling**[1], **Jean Yang**[3,4], **Anthony Liu**[1,5], **Ralph Nanan**[1,5]*

**1** Sydney Medical School–Nepean, Discipline of Paediatrics and Child Health, The University of Sydney, Sydney, NSW, Australia, **2** Centre for Personalized Medicine, Linköping University, Linköping, Sweden, **3** School of Mathematics and Statistics, The University of Sydney, Sydney, Australia, **4** The Judith and David Coffey Life Lab, Charles Perkins Centre, University of Sydney, Sydney, Australia, **5** Charles Perkins Centre–Nepean, The University of Sydney, Sydney, Australia

* ralph.nanan@sydney.edu.au

## Abstract

Asthma is the most common non-communicable pulmonary condition, affecting prepubertal boys more often than girls. This study explored how maternal and perinatal risk factors are linked to poorly controlled childhood asthma in a sex dependent manner. This single centre study was performed at a metropolitan teaching hospital in Western Sydney, Australia, using electronical obstetric records from 2000 to 2017 and electronical pediatric records from 2007 to 2018. The data of 1694 children with complete entries were retrospectively analysed. Risk factors for multiple hospital admission for asthma were selected by backward-eliminated Poisson regression modelling. Selection stability of these parameters was independently confirmed using approximated exhaustive search. Sex-specific regression models indicated that most notably parity (RR[95%CI] for parity = 3; 1.85[1.22–2.81]), birth length z-score (1.45[1.23–1.70]) and birth weight z-score (0.77[0.65–0.90]) contributed to multiple asthma admissions in girls, while boys were affected most prominently by maternal BMI (e.g. BMI 35–39.9; 1.92[1.38–2.67]) and threatened preterm labor (1.68[1.10–2.58]). Allergic status was a risk factors for both boys and girls (1.47[1.18–1.83] and 1.46[1.13–1.89]). Applying ROC analysis, the predictive modelling of risk factors for hospital admissions showed an incremental increase with an AUC of 0.84 and 0.75 for girls and boys respectively for >3 hospital admissions. Multiple hospital admissions for asthma are associated with maternal and perinatal risk factors in a sex and birth order dependent manner. Hence, prospective risk stratification studies aiming to improve childhood asthma control are warranted to test the clinical utility of these parameters. Furthermore, the influence of the early in utero environment on male-female differences in other communicable and non-communicable respiratory conditions should be considered.

**Data Availability Statement:** The non-anonymized data set cannot be shared publicly in order to protect patient integrity. Requests to access the non-anonymized dataset can be made to the

Nepean Mountains Local Health District (Human Research Ethics Office, Nepean Hospital, PO Box 63, Penrith NSW 2751, Australia; Phone: +6124734 3441; Fax: +6124734 1967; Email: NBMLHD-Ethics@health.nsw.gov.au). A minimal, anonymized version of the data of included patients can be found in the Supporting Information files.

**Funding:** The author(s) received no specific funding for this work.

**Competing interests:** The authors have declared that no competing interests exist.

## Introduction

Asthma is a chronic disease affecting approximately 339 million people globally [1], making it one of the most common non-communicable diseases in the world. In terms of disability and premature death, the pediatric population carries the greatest burden of asthmatic disease [1]. While deaths due to asthma have decreased steadily, prevalence has increased by 17% between 2006 and 2016 [1]. Genetic changes are too slow to account for this abrupt rise in prevalence and perinatal factors have thus gained more interest [2]. This concept is a focus of the developmental origins of health and disease (DOHaD) hypothesis [3].

The DOHaD hypothesis proposes that epigenetic factors induced by the *in utero* environment might be part of evolutionary adaptive strategies to fit predicted postnatal environments [3]. If there is a mismatch between the *in utero* predicted and postnatally encountered environment the infant may not be able to adapt to these changes, which has been associated with increased risk for chronic disease later in life [4,5]. It was, for example, recently shown that perinatal stress could have a life-long impact on the hypothalamic-pituitary-adrenal axis, resulting in decreased CD8+ T cell function and hence long-term immunological consequences [6].

Asthma also appears to be initiated *in utero* [7] with increasing evidence supporting an association of development of asthma in childhood and a wide range of perinatal factors [2,7,8]. While one set of studies indicate that asthma diagnosis might be associated with parity, low birth weight, caesarian section, maternal age at delivery, maternal asthma, socioeconomic status and maternal smoking, another set of independent studies report a strong association with male sex [7,9–11]. In this regard it has for example been shown that prepubertal boys have increased allergic inflammation and serum IgE levels as well as narrower airways relative to lung volume compared to girls which could make them more likely to exhibit symptoms [12]. How these sexual differences are mediated to begin with can though only be speculated on in these association studies.

Animal models indicate that environmental insults during pregnancy program male and female fetuses differently [13,14]. Human studies suggest that epigenetic differences between males are more pronounced than between females which could suggest that males are more sensitive to environmental factors than females [15]. In order to harness the concept of DOHaD for clinical purposes, we here aim to examine sex-dependent differences in the impact of maternal and perinatal factors on poorly controlled childhood asthma. Insights might be used to develop clinical strategies capable of stratifing patients by risk group, and hence allowing for early diagnosis and adequate preventative resource allocation that can prevent hospital admission and lessen the burden of disease.

## Materials and methods

### Study design

In this retrospective cohort study, the association of maternal and perinatal factors, and childhood asthma was examined, analysing electronic paediatric and obstetric hospital records. This single centre study was performed at a metropolitan teaching hospital in Western Sydney, New South Wales, Australia. Data was retrieved from the electronic databases of the maternity and paediatric departments.

### Study population

The obstetric database (eMaternity) is a state-wide mandatory obstetric database and reported on 67268 babies delivered between January 2000 and December 2017. Patients from this data

base were included in the study if they had a unique medical record number (MRN) and if their sex was known. Entries from this register were excluded if a stillbirth was reported or if maternal parity exceeded 15. Hence 50352 patients were included. The pediatric electronic database between from January 2007 and February 2018 and reported on 7783 patients that had been admitted to the pediatric ward 10261 times collectively. Patients were included if their allergy status, sex and weight was recorded at admission to the pediatric ward (n = 7335). Collectively 9768 admissions were reported for the 7335 patients, multiple admissions of the same patient were detected by this patients MRN and data from all entries was aggregated. Records from the paediatric database could be linked to eMaternity based on each patient's unique MRN. This occurred for 1694 of the included patients.

## Data measurements

Information included in eMaternity concerned maternal and perinatal outcomes routinely documented in the context of pregnancy, such as infant parameters, pregnancy associated variables and maternal health information. A full list of the recorded variables as well as their definitions can be found in **S1 Table**. From the pediatric data base information on the patient's sex, weight, length, head circumference, allergies, admission age, diagnoses, allergy status, medical history and diagnosis was extracted. Childhood allergy was defined as parental reported allergy for any agent.

The pediatric data base was used to ascertain admission and asthma diagnosis. Asthma admissions were defined as physician diagnosed asthma requiring admission to a paediatric ward. Multiple admissions for asthma were used as a surrogate for poorly controlled asthma. This is based on a uniform approach for asthma management applying the NSW Asthma guidelines [16]. In this context, admission criteria are inability to stretch reliever therapy to third hourly inhaled salbutamol following acute asthma management [16]. As it can be difficult to diagnose asthma before the age of two years [17], we excluded all patients below this age except for 98 patients aged 18–23 months who had a solid diagnosis based on strong family history of asthma, multiple hospital admissions for asthma or being on an asthma medication.

## Statistical analysis

**Data management.** Observations for arterial cord blood pH, lactate and base excess beyond four standard deviations were considered outliers and removed, thereafter mean and SD were recalculated. BMI was calculated as a function of the mother's pre-pregnancy weight and height. BMI was then categorized in compliance with current BMI categories by the World Health Organization.

Perinatal growth curves were created based on the anthropometric measurements of all 50352 obstetric patients that passed inclusion criteria (**S1 File**). Growth curves were used to convert birth weight, length and head circumference (HC) into gestational age and sex adjusted z-scores which made anthropometric measurements comparable between children of different gestational age and sex.

**Population characteristics.** Standard descriptive statistics were applied during the creation of the population characteristics. Continuous variables were analysed using t-test or Mann-Whitney U-test. Categorical variables were analysed using chi-square test. Significance level was defined as $P < 0.05$.

**Variable selection.** Initially 23 variables (shown in **Table 1**) entered a Poisson regression model with asthma admittance frequency as response variable, stepwise backward elimination was applied using the Akaike Information Criterion (AIC) as the stopping criterion. The predictors remaining significant were used to fit separate sex-specific Poisson models. In order to

**Table 1. Comparison of included and excluded population characteristics.**

| Obstetric characteristics | | | | | | | |
|---|---|---|---|---|---|---|---|
| | Included (n = 1694) | Excluded (n = 48658) | | | Included (n = 1694) | Excluded (n = 48658) | |
| Variables | % | % | P | Variables | % | % | P |
| **Parity** | | | $<10^{-8}$ | **Smoking** | | | 0.046 |
| 0 | 44.27% | 37.77% | | No | 80.87% | 76.88% | |
| 1 | 30.28% | 30.85% | | Yes | 19.13% | 20.65% | |
| 2 | 15.11% | 16.85% | | Not specified | 0.00% | 2.47% | |
| 3 | 6.49% | 7.70% | | **Alcohol** | | | 0.531 |
| ≥3 | 3.84 | 6.83 | | No | 97.70% | 95.56% | |
| **Conception mode** | | | 0.726 | Yes | 0.30% | 0.42% | |
| Spontaneous | 96.16% | 93.47% | | Not specified | 2.00% | 4.03% | |
| Assisted | 3.84% | 3.93% | | **Illegal drugs** | | | 0.273 |
| Not specified | 0.00% | 2.59% | | No | 98.11% | 95.14% | |
| **TPL** | | | 0.072 | Yes | 1.89% | 2.37% | |
| No | 96.22% | 94.70% | | Not specified | 0.00% | 2.59% | |
| Yes | 3.78% | 4.73% | | **Infant sex** | | | $<10^{-3}$ |
| Not specified | 0.00% | 0.57% | | Female | 43.80% | 48.51% | |
| **Diabetes** | | | 0.288 | Male | 56.20% | 51.49% | |
| No | 92.74% | 89.91% | | **Mode of birth** | | | 0.005 |
| Gestational | 6.26% | 7.06% | | Vaginal | 62.63% | 65.91% | |
| Pre-existing | 1.00% | 0.86% | | CS | 37.37% | 34.09 | |
| Not specified | 0.00% | 2.18% | | **Feeding discharge** | | | 0.515 |
| **Season of birth** | | | 0.357 | Infant formula | 23.44% | 22.48% | |
| Spring | 23.20% | 25.07% | | Breast milk | 76.56% | 76.40% | |
| Summer | 24.85% | 24.08% | | Not specified | 0.00% | 1.12% | |
| Autumn | 25.44% | 25.25% | | **Maternal age** | | | 0.384 |
| Winter | 26.51% | 25.60% | | < 20 | 5.61% | 6.07% | |
| **BMI** | | | 0.629 | 20 to 35 | 80.28% | 78.90% | |
| Underweight | 11.39% | 10.42% | | > 35 | 14.11% | 15.03% | |
| Normal | 40.38% | 40.16% | | | | | |
| Overweight | 24.77% | 22.98% | | | Mean±SD | Mean±SD | P |
| Obese class I | 13.58 | 13.29% | | **Birth weight (z)** | 0.036±0.95 | -0.001±1.00 | 0.111 |
| Obese class II | 7.03% | 6.34% | | **Birth length (z)** | 0.048±0.97 | -0.002±1.00 | 0.038 |
| Obese class III | 4.13% | 4.73% | | **Birth HC (z)** | 0.070±0.99 | -0.001±1.00 | 0.003 |
| Not specified | 0.00% | 2.08% | | **Gestational age** | 39.32±1.67 | 38.92±2.50 | $<10^{-19}$ |
| **APH after 20weeks** | | | 0.005 | **ACB pH** | 7.28±0.07 | 7.27±0.08 | $<10^{-3}$ |
| No | 97.46% | 96.10% | | **ACB BE** | -1.08±3.65 | -1.74±3.85 | $<10^{-12}$ |
| Yes | 2.54% | 3.91% | | **ACB lactate** | 4.02±1.69 | 4.03±1.96 | 0.907 |
| Pediatric characteristics | | | | | | | |
| | Included (n = 1694) | Excluded (n = 5641) | | | Included (n = 1694) | Excluded (n = 5641) | |
| Variables | % | % | P | Variables | Median [IQR] | Median [IQR] | P |
| **Allergies** | | | 0.035 | **n Asthma admissions** | 0 [0–0] | 0 [0–0] | $<10^{-5}$ * |
| No | 74.62% | 77.13% | | **Admission age (y)** | 3.66 [2.64–5.35] | 6.18 [3.54–10.67] | $<10^{-147}$ |
| Yes | 25.38% | 22.87% | | | | | |
| Not specified (n) | 0 | 519 | | | | | |

Abbreviations: TPL, threatened premature labor; BMI, body mass index; APH, antepartum premature hemorrhage; CS, caesarian section; HC, head circumference; ACB, arterial cord blood; BE, base excess; *, less asthma admissions in excluded group (mean included = 0.37, mean excluded = 0.26).

further investigate the variable selection stability of our backward eliminated Poisson regression model, we used the methods described in [18].

## Ethics

This study complies with the Helsinki Declaration and relevant Australian guidelines and was approved by the Human Research Ethics Committee of the Nepean Blue Mountain Local Health District (Ethics number: 10/16). The accessed data were not anonymized. As this was a retrospective study the need for patient consent was not required according to the ethics committee.

## Results

### Population characteristics

Data was derived from an obstetric and a pediatric database. After inclusion and exclusion criteria were applied to the records of respective databases the obstetric data base documented the birth related parameters of 50352 children. The pediatric data documented 9768 admissions of 7335 children to the pediatric ward (**Fig 1**). The information of multiple admissions of a unique patient to the pediatric ward was aggregated and information on the number of

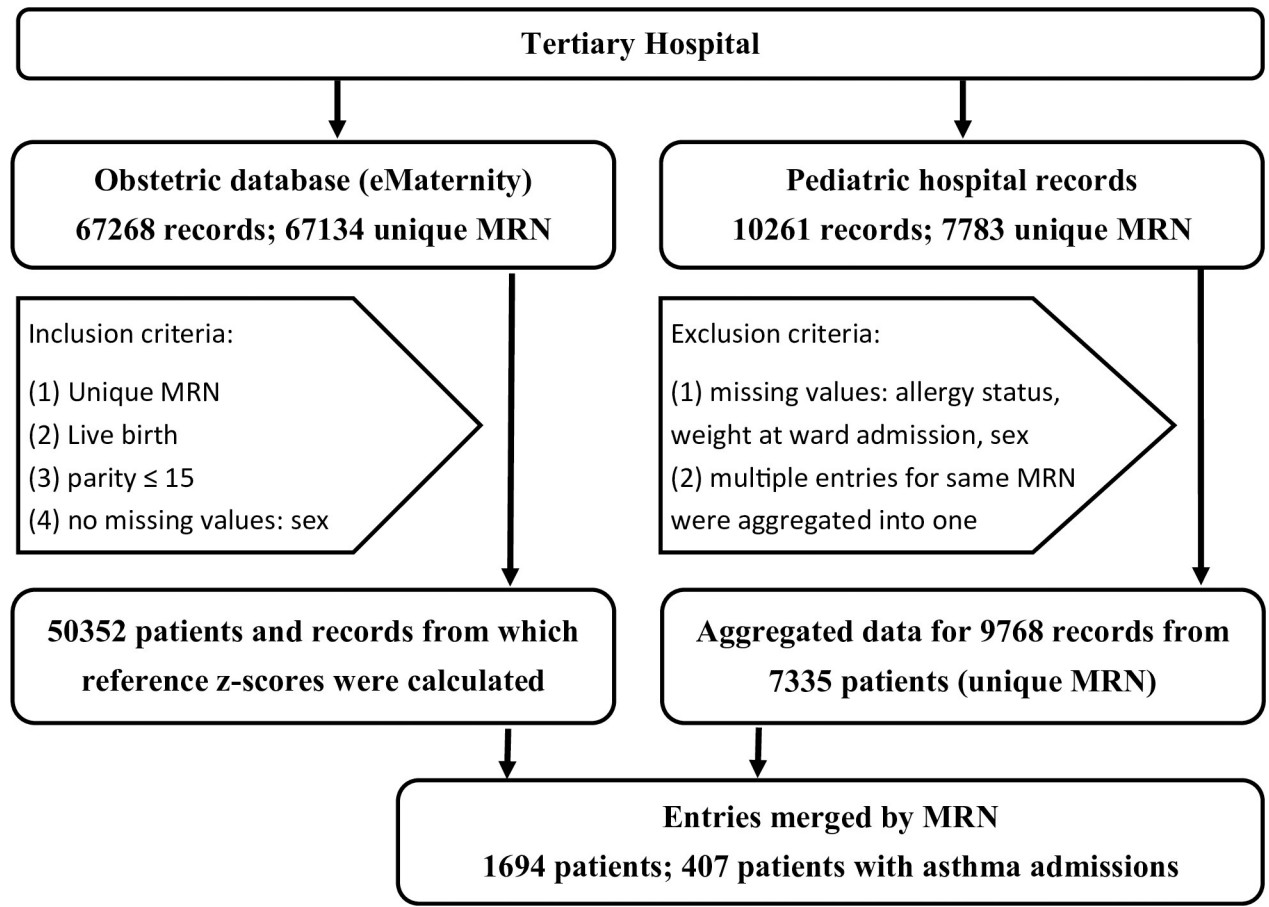

**Fig 1. Flow chart describing inclusion and exclusion criteria as well as database merging.** Main results from this article are derived from 1694 patients for which both paediatric hospital records and obstetric records were available. Reference z-scores for perinatal growth curves were derived from a subset of the obstetric database, using the inclusion criteria indicated. Abbreviation: MRN, medical record number.

asthma related admissions was extracted. Entries from both databases were matched using the patient specific medical record number, which resulted in the inclusion of 1694 patients (407 with asthma admissions).

To ensure the representability of our selection we compared the included with excluded children of the same database (Table 1). Differences between groups, such as included children having a lower parity, being more frequently male, being born at a lower gestational age and more allergies, reflect general risk factors for increased prevalence of paediatric disease [19]. Interestingly we also notice that excluded children are older than included children, which is likely due to exclusion of children without a paediatric record (dated back to 2007), which excluded many children in eMaternity born prior to 2007. When stratifying the included children by asthma admission frequency, age distributions of included children were similar although children with more asthma admissions naturally tended to be slightly older (S1 Fig).

To avoid multicollinearity between gestational age and anthropometric parameters in our regression models we created perinatal growth curves (S1 File) based on which z-scores for the birth weight, length and HC of all 1694 children could be calculated (S2–S7 Figs).

### Risk factors for multiple hospitalizations due to asthma

Following backward elimination, eight of 23 variables remained in a Poisson regression model (AIC = 2936.429; p < $10^{-9}$; Table 2) for asthma admittance frequency. To test for sex-specific effects, these eight variables were then entered in Poisson regression models for males and females respectively without applying any backward elimination (Table 2). Looking at both sexes, multiple hospital admission for asthma was associated with allergies, parity, maternal pre-pregnancy BMI, threatened premature labour (TPL), birth weight, birth length, feeding method (breastfed vs. formula fed; P = 0.08), as well as the child's sex (P = 0.059). Higher birth

**Table 2. Poisson regression models for multiple hospital admission for asthma.**

| Variables | Male & Female | | Female | | Male | |
|---|---|---|---|---|---|---|
| | RR (95% CI) | P | RR (95% CI) | P | RR (95% CI) | P |
| Allergic | **1.46 (1.24–1.73)** | **<$10^{-5}$** | **1.46 (1.13–1.89)** | **<$10^{-2}$** | **1.47 (1.18–1.83)** | **<$10^{-3}$** |
| z-score birth length | **1.22 (1.10–1.35)** | **<$10^{-3}$** | **1.45 (1.23–1.70)** | **<$10^{-5}$** | 1.08 (0.95–1.23) | 0.26 |
| BMI:underweight | 0.79 (0.59–1.06) | 0.11 | 1.06 (0.69–1.64) | 0.78 | **0.62 (0.42–0.92)** | **0.02** |
| BMI:pre-obese | 1.03 (0.84–1.27) | 0.75 | 1.21 (0.88–1.67) | 0.24 | 0.93 (0.71–1.23) | 0.61 |
| BMI:obese class I | 1.12 (0.88–1.42) | 0.38 | 1.34 (0.91–1.96) | 0.14 | 0.98 (0.71–1.33) | 0.88 |
| BMI:obese class II | **1.65 (1.26–2.16)** | **<$10^{-3}$** | 1.29 (0.81–2.06) | 0.29 | **1.92 (1.38–2.67)** | **<$10^{-3}$** |
| BMI:obese class III | 1.03 (0.67–1.57) | 0.90 | 0.86 (0.41–1.78) | 0.68 | 1.13 (0.67–1.90) | 0.65 |
| TPL: yes | **1.65 (1.18–2.30)** | **<$10^{-2}$** | 1.47 (0.84–2.56) | 0.18 | **1.68 (1.10–2.58)** | **0.02** |
| Parity: 1 | 0.89 (0.74–1.08) | 0.24 | **0.71 (0.52–0.97)** | **0.03** | 1.04 (0.82–1.33) | 0.73 |
| Parity: 2 | 0.89 (0.70–1.14) | 0.36 | 0.76 (0.51–1.12) | 0.16 | 1.00 (0.72–1.37) | 0.99 |
| Parity: 3 | **1.49 (1.13–1.97)** | **<$10^{-2}$** | **1.85 (1.22–2.81)** | **<$10^{-2}$** | **1.31 (0.90–1.92)** | **<$10^{-3}$** |
| Parity: > 3 | 1.06 (0.72–1.57) | 0.77 | 0.98 (0.51–1.88) | 0.96 | 1.10 (0.67–1.80) | 0.70 |
| z-score birth weight | **0.87 (0.78–0.97)** | **0.02** | **0.77 (0.65–0.90)** | **<$10^{-2}$** | 0.97 (0.84–1.12) | 0.67 |
| Sex: Male | 1.17 (0.99–1.37) | 0.06 | - | - | - | - |
| Breastfed | 1.19 (0.98–1.45) | 0.08 | 1.11 (0.82–1.50) | 0.51 | 1.28 (0.99–1.66) | 0.06 |
| | **AIC = 2936.429, P < $10^{-9}$** | | **AIC = 1225.457, P < $10^{-5}$** | | **AIC = 1722.212, P < $10^{-5}$** | |

Abbreviations: TPL, threatened preterm labor, AIC, Akaike information criterion, RR, risk ratio.

weight was inversely associated with hospital admissions for asthma. While categories such as maternal BMI obese class III as well as parity>3 were included in the model they did not reach statistical significance, likely due to the few individuals included in these categories (**Table 1**). In females, birth weight, birth length as well as being born second (parity = 1), compared to firstborn, was associated with multiple asthma admission. This is an interesting finding as those risk factors were not seen in males. The two risk factors affecting multiple asthma admissions most drastically in the combined model, namely TPL and obesity class II, were not significant in the female model.

Conversely, TPL and maternal pre-pregnancy obesity class II, were the strongest risk factors for multiple asthma admissions in males, while birth weight and length did not seem to affect the hospitalization risk. Additionally, maternal pre-pregnancy underweight seemed to have a rather strong inverse association with asthma related hospitalization in males and, though only borderline significant (P = 0.06), it seemed that feeding method affected the risk of hospital in males, while not at all having an effect on females.

This sex-based stratification of our population revealed that few of the eight risk factors, that were found to influence multiple hospital admissions due to asthma generally, effected both males and females. Indeed, only two of these eight risk factors, namely parity and allergy status, were shared between males and females. However, parity seemed to effect females to a greater extent than males (**Table 2**).

The potential of the eight risk factors for stratifying asthmatic patients into low-risk and high-risk groups is further demonstrated in the ROC-curves (**Fig 2**). In particular, we find that the above regression model can differentiate quite accurately between patients being hospitalized zero to three times and patients being hospitalized four or more times. Furthermore, the statistical stability of risk factors chosen by the backward eliminated Poisson regression model was demonstrated by application of APproximated Exhaustive Search (APES; **S8**–**S10 Figs**). APES indicated that risk factors included in the above Poisson regression models, indeed were frequently chosen in repeated bootstrap subsampling which is indicative of the generalizability of our findings. All eight risk factors selected in the general Poisson regression model were also selected in the general APES model, furthermore, APES indicated that gestational and pre-existing diabetes, season of birth, gestational age, caesarean section and arterial cord blood pH might have additional predictive value. Likewise, the general APES model, sex-specific APES models were in reasonably good agreement with the sex-specific Poisson models.

## Discussion

Similar to other lung-diseases [20,21] and previous findings in asthma [10], we find that hospital admission due to asthma is sex-biased. This raised the question if asthma develops differently in males and females due to differences in perinatal factors. This retrospective study therefore combined paediatric and obstetric data to investigate sex-specific patterns in the effect of perinatal factors on multiple asthma admissions in children.

A Poisson regression including both sexes found a number of perinatal factors to influence risk for multiple asthma admissions. In accordance with previous research [22,23], we found that allergic status was associated with multiple asthma admissions. The effect of allergic status seems uniform among sexes. Also, in agreement with previous studies [7,9–11,19], male sex (P = 0.06) was found to possibly influence risk for multiple hospital admissions, to investigate the relationship between sex and asthma further we conducted additional Poisson regressions were implemented for males and females respectively. Most significant in females, high parity, high birth length and low birth weight contributed to multiple asthma admissions in childhood, while being second born (parity = 1) decreased the risk. Conversely, in males a high

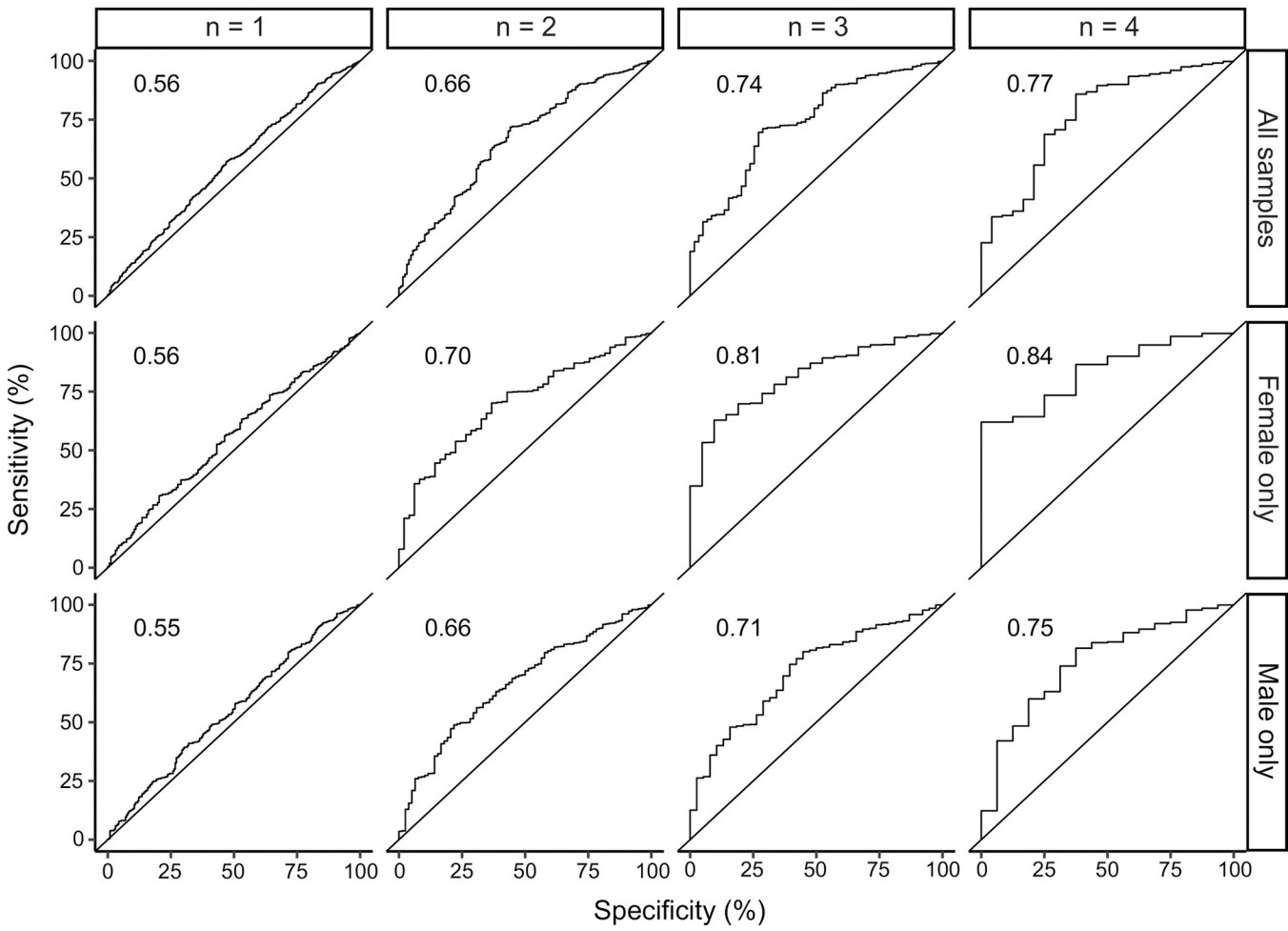

**Fig 2. ROC-curves showing prediction accuracy of the Poisson regression model for classifying samples with more than n visits.** Each column depictures the accuracy of the Poisson model classifying patients into having more than n hospital admissions or less than/equal with n hospital admissions. As shown, prediction accuracy is highest when n ≥ 3 and tends to be slightly higher in females than in males. AUC is annotated within each graph.

maternal BMI and TPL were the most significant risk factors for increased multiple asthma admissions. This suggests that risk factors in the multiple asthma admissions have a sex-specific pattern.

## Male risk factors

In this study we find that children born to obese mothers (BMI>35kg/m$^2$) are at greater risk for multiple asthma admissions. This coincides well with previous findings linking increasing maternal weight to asthma in the offspring [24–26]. Other studies suggest that this association holds true even when accounting for gestational weight gain [24,26] and the child's own weight [25]. To our knowledge, our study is first to show that maternal obesity is associated with multiple asthma admissions in male, but not female, offspring. That the obesity class III (BMI>40kg/m$^2$) did not reach statistical significance is probably due to insufficient sample size. It is likely that the risk for multiple hospital admissions for asthma continues to increase with increasing BMI.

Previous studies have found that obese women's placentas are more commonly dysfunctional, showing decreased oxygen and nutrient delivery [27]; this obesogenic intrauterine

environment could alter foetal lung development through oxidative stress [27] and dysregulation of the maternal hypothalamic-pituitary-adrenal-axis [28]. Giussani *et al.* showed in sheep, that male foetuses had a twofold greater increase in plasma cortisol in response to hypoxia compared to female foetuses [29]. Obesity is also a state of chronic low-grade inflammation where adipose tissue expresses increased levels of leptin and decreased levels of adiponectin [30], which have been shown to effect foetal lung development and airway reactivity [30,31]. Our finding that only males were subject to risk increase through maternal BMI could therefore potentially be explained by males' apparent susceptibility to inflammatory and hypoxic stress *in utero*.

We also found an association between TPL and asthma admissions in male offspring. As gestational age was excluded from the general model this is unlikely reflecting the increased risk of premature birth for male foetuses [32]. Instead, it is likely to reflect that medications used for treatment of TPL can alter the intrauterine environment and thereby affect foetal lung development [33]. It might also be that increase of asthma severity could be connected to the risk factors underlying TPL that are not included in the model, such as immunological factors [34] and intrauterine infection [34].

### Female risk factors

We found that a parity = 3 was associated with an increased risk for multiple hospital admissions for asthma in both males and females, though having the strongest effect on females. Furthermore, it seemed that female second-borns (parity = 1) had a lower risk for asthma related hospitalization, compared to a firstborns (parity = 0). Even previous studies have found birth order to influence asthma severity, though results are ambiguous. Some studies report that having many older siblings increases the severity of asthma [35], while others claim the opposite [36]. In one recent study, Kikkawa *et al.* found that the risk for asthma increases with birth order in early childhood, but after 10 years of age the risk for asthma decreases with birth order [37].

In our Poisson regression, the risk for severe asthma was affected by birth weight (adjusted for gestational age). This agrees well with previous reports [2,7,8]. Interestingly, we find that risk for hospital admission for asthma only was increased in small for gestational age females and not males. Similarly, birth length (adjusted for gestational age) seemed to only affect the risk for severe asthma in females. While the exact mechanism behind this sex-biased association of foetal growth and the number of hospital admissions for asthma warrants further research, it might be speculated that sex-dependent differences in the foetal-maternal intrauterine communication could be involved. Further, it cannot be ruled out that this outcome is biased by uncontrolled maternal asthma, which has been reported to be associated with reduced female foetal growth specifically [38].

### Strengths and limitations

A strength of this study is the validation of the backward eliminated variable selection in the Poisson regression by the new statistical approach APES [18]. The retrospective study design allowed us to account for many variables, did however not allow us to account for family history of asthma or socioeconomic status, which are known risk factors for asthma [2,7,9]. It is also worth mentioning that this is a monocentric study and bigger longitudinal studies are needed to investigate the possibility of sex-specific risk factors in the severity of childhood asthma. The interpretation of the here presented results is further complicated as multiple hospital admission for asthma could have intrinsic causes, as assumed here, but also extrinsic causes in the form of non-compliance.

## Conclusion

As shown in this study, environmental influences during pregnancy are likely to result in different foetal programming in males and females. Maternal obesity for example is associated with increasing number of hospital admissions for asthma in male, but not female, offspring. These sex-specific patterns in asthma severity warrant further mechanistical and epidemiological investigation and could have profound implications in aiming prevention efforts as well as in treatment strategies. In a broader context, evidence for the importance of the perinatal environment for later-life health is increasing and it will be imperative for future research to consider whether inherent sex-differences could originate *in utero*. Especially future research should consider sex when assessing risk factors, symptoms, and treatment outcomes for asthma.

## Supporting information

**S1 Fig. Included children stratified by admission age and asthma admission frequency.** (PDF)

**S2 Fig. Growth curves for birth weight by gestational age, starting at 22 weeks gestation.** Children below 32 weeks gestation were excluded from our study population, but as growth references for small gestational ages are scarce growth curves for future reference were created starting at 22 weeks gestation. One dot represents one child. (PDF)

**S3 Fig. Growth curves for birth length by gestational age, starting at 22 weeks gestation.** Children below 32 weeks gestation were excluded from our study population, but as growth references for small gestational ages are scarce growth curves for future reference were created starting at 22 weeks gestation. One dot represents one child. (PDF)

**S4 Fig. Growth curves for birth head circumference by gestational age, starting at 22 weeks gestation.** Children below 32 weeks gestation were excluded from our study population, but as growth references for small gestational ages are scarce growth curves for future reference were created starting at 22 weeks gestation. One dot represents one child. (PDF)

**S5 Fig. Growth curves for birth weight from gestational age 32 upward.** (PDF)

**S6 Fig. Growth curves for birth length from gestational age 32 upward.** (PDF)

**S7 Fig. Growth curves for birth head circumference from gestational age 32 upward.** (PDF)

**S8 Fig. Variable selection for prediction of number of asthma related hospitalization in the general population, using APES.** Included in the plot is the AIC and BIC threshold, which both present penalized-likelihood criteria which can be used to choose the best predictor subset. AIC, Akaike information criterion; BIC, Bayesian information criterion. (PDF)

**S9 Fig. Variable selection for prediction of number of asthma related hospitalization in the female population, using APES.** Included in the plot is the AIC and BIC threshold, which both present penalized-likelihood criteria which can be used to choose the best predictor

subset. AIC, Akaike information criterion; BIC, Bayesian information criterion.
(PDF)

**S10 Fig. Variable selection for prediction of number of asthma related hospitalization in the male population, using APES.** Included in the plot is the AIC and BIC threshold, which both present penalized-likelihood criteria which can be used to choose the best predictor subset. AIC, Akaike information criterion; BIC, Bayesian information criterion.
(PDF)

**S1 Data. Minimal, anonymized data for included patients.**
(PDF)

**S1 File. Supplementary information and methods.** This file includes supplementary method descriptions, supplementary figure descriptions as well as supplementary tables.
(DOCX)

## Author Contributions

**Conceptualization:** Samuel Schäfer, Felicia Sundling, Anthony Liu, Ralph Nanan.

**Data curation:** Samuel Schäfer, Felicia Sundling, Anthony Liu.

**Formal analysis:** Samuel Schäfer, Kevin Wang, Felicia Sundling.

**Investigation:** Samuel Schäfer.

**Methodology:** Samuel Schäfer, Kevin Wang, Felicia Sundling, Jean Yang, Anthony Liu, Ralph Nanan.

**Project administration:** Samuel Schäfer, Jean Yang, Anthony Liu, Ralph Nanan.

**Software:** Kevin Wang.

**Supervision:** Jean Yang, Anthony Liu, Ralph Nanan.

**Visualization:** Samuel Schäfer, Kevin Wang.

**Writing – original draft:** Samuel Schäfer, Felicia Sundling.

**Writing – review & editing:** Samuel Schäfer, Kevin Wang, Jean Yang, Anthony Liu, Ralph Nanan.

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
