## [Decision Letter · Decision Letter 0]

2 Mar 2021

PONE-D-21-03952

Modelling Maternal and Perinatal Risk Factors to Predict Poorly Controlled Childhood Asthma

PLOS ONE

Dear Dr. Ralph Nanan,

Thank you for submitting your manuscript to PLOS ONE. After careful consideration, we feel that it has merit but does not fully meet PLOS ONE’s publication criteria as it currently stands. Therefore, we invite you to submit a revised version of the manuscript that addresses the points raised during the review process.

We look forward to receiving your revised manuscript.

Kind regards,

Kazumichi Fujioka

Academic Editor

PLOS ONE

Journal Requirements:

2.Please provide additional details regarding participant consent. In the ethics statement in the Methods and online submission information, please ensure that you have specified (1) whether consent was informed and (2) what type you obtained (for instance, written or verbal, and if verbal, how it was documented and witnessed). If your study included minors, state whether you obtained consent from parents or guardians. If the need for consent was waived by the ethics committee, please include this information.

3.We note that you have indicated that data from this study are available upon request. PLOS only allows data to be available upon request if there are legal or ethical restrictions on sharing data publicly. For information on unacceptable data access restrictions, please see http://journals.plos.org/plosone/s/data-availability#loc-unacceptable-data-access-restrictions.

Reviewers' comments:

Reviewer's Responses to Questions

**Comments to the Author**

1. Is the manuscript technically sound, and do the data support the conclusions?

Reviewer #1: Partly

Reviewer #2: Yes

2. Has the statistical analysis been performed appropriately and rigorously? 

Reviewer #1: Yes

Reviewer #2: Yes

3. Have the authors made all data underlying the findings in their manuscript fully available?

Reviewer #1: Yes

Reviewer #2: No

4. Is the manuscript presented in an intelligible fashion and written in standard English?

Reviewer #1: Yes

Reviewer #2: Yes

5. Review Comments to the Author

Reviewer #1: General Comments:

This study showed that maternal obesity and threatened premature labour (TPL) are associated with the development of asthma in children. This is consistent with recent findings that epigenetics is involved in the development of asthma. The number of participants in the study is large, the missing data is small, and the value of the data is very high. However, it is difficult to understand the intent of the study and I suggest a revision of the description and analysis.

Specific recommendations for revision major:

You should emphasize more in the introduction why you have analyzed the subject separately as male and female. Revising the following four points will make the reason more clear.

1. line66-69

Furthermore, male sex has been linked to a more serious asthma phenotype during childhood with increased risk of hospitalization (9, 12). This is in agreement with sex-biases seen in other lung diseases such as influenza (13) and COVID-19 (14) and might relate to fundamental sex-differences that are minted in utero.

It is true that boys are more likely to develop asthma in childhood than girls. However, I feel that this fact is an insufficient reason for this study to be analyzed separately for boys and girls. I suggest that this part be deleted.

2. line 70-71

Animal models indicate that environmental insults during pregnancy program male and female fetuses differently (15, 16), though it is unclear to what extent this applies to humans.

This is probably the reason why this study was conducted separately for males and females. It is known that males are more prone to epigenetic mutations caused by environmental factors than females. Didn't you assume that it would be easier to identify the environmental factors that lead to epigenetic variation if you analyzed only males?

Not only in animal models, but also in human studies, it has been found that males are more prone to epigenetic mutations caused by environmental factors than females. I would suggest adding the following references for example.

Within-pair differences of DNA methylation levels between monozygotic twins are different between male and female pairs

BMC Med Genomics. 2016; 9: 55.

https://pubmed.ncbi.nlm.nih.gov/27561550/

3. line 166-167

Following backward elimination, eight of 23 variables remained in the Poisson regression model (AIC = 2936.429; p < 10-9 ; Table 2).

Based on the hypothesis that men are more strongly affected by epigenetics, it would be better to use only men in the selection of risk factors. I would like you to discuss this with the statisticians.

Furthermore, it is puzzling that smoking during pregnancy was not included in the eight risk factors selected. This problem will be solved if smoking is found to be associated with asthma by analysis of males only.

4. line 262-264

Mechanistically, it is possible that the intrauterine environment is permanently altered through pregnancies and that these alterations influence female foetuses to a greater extent than male foetuses.

This speculation contradicts the hypothesis that men are more prone to epigenetic mutations caused by environmental factors than women. I feel that it is unnecessary to discuss female risk factors in order to clarify the intent of this study.

Specific recommendations for revision minor:

1. line 248-250

Instead, it is imaginable that medications used to treat TPL could alter the intrauterine environment and thereby affect foetal lung development.

I propose to add the following paper.

Beta-2 receptor agonist exposure in the uterus associated with subsequent risk of childhood asthma

Pediatr Allergy Immunol. 2017; 28: 746-753.

https://pubmed.ncbi.nlm.nih.gov/28892561/

Reviewer #2: Interesting subject and good study. The key message here are sex differences in asthma admission rate and prenatal risk factors.

Minor remarks:

Abstract - statement “condition affecting boys more often than girls” – is not accurate while it’s true only before puberty – see ref. 10, and quiet different later in life when women have more asthma than men. I will be helpful to add on some information form literature why those difference are observed - e.g. lung development, total IgE level.

I suggest to add an information about the asthma admission cases – what is the age, sex and how many admissions. As I believe it’s 17 years period of time, so we have both 2 y. and 17 y. children?

Since asthma is a different phenotype before 5y. and after – it’s important to differentiate according to age. So called “childhood asthma” is triggered by infections, which are more common earlier in life in second child, than in first, and some young children outgrow symptoms later in life.

Socioeconomic status is an important risk factor affecting asthma admission related mainly to regularity of treatment but also smoking and obesity. The lack of this data is a disadvantage of this study, however possibly doesn’t affect the sex differences which were observed.

In table 2 the data regarding admission is missing? - 0-0? Or not clear.

6. PLOS authors have the option to publish the peer review history of their article (what does this mean?). If published, this will include your full peer review and any attached files.

Reviewer #1: **Yes: **Mitsuhiro Okamoto

Reviewer #2: No

---

## [Author Response · Author response to Decision Letter 0]

26 Apr 2021

PONE-D-21-03952

Modelling Maternal and Perinatal Risk Factors to Predict Poorly Controlled Childhood Asthma

Dear Editor,

We resubmit the manuscript after having implemented all by the reviewers suggested changes. We have also ensured that the manuscript meets PLOS ONE style requirements and additional detail on participant consent has been added.

We took also note the journals requirement of data availability. As this study was conducted on non-anonymized patient records. We cannot share the non-anonymized data publicly. This is to protect patient integrity. Requests to access the non-anonymized dataset can be made to the Nepean Mountains Local Health District and this information has been added to the text. However, we have uploaded a minimal, anonymized version of the data of included patients as Supporting Information.

Kind regards,

Ralph Nanan, on behalf of all authors

Response to reviewers' comments:

Reviewer #1: General Comments:

This study showed that maternal obesity and threatened premature labour (TPL) are associated with the development of asthma in children. This is consistent with recent findings that epigenetics is involved in the development of asthma. The number of participants in the study is large, the missing data is small, and the value of the data is very high. However, it is difficult to understand the intent of the study and I suggest a revision of the description and analysis.

Specific recommendations for revision major:

You should emphasize more in the introduction why you have analyzed the subject separately as male and female. Revising the following four points will make the reason more clear.

1. line66-69

Furthermore, male sex has been linked to a more serious asthma phenotype during childhood with increased risk of hospitalization (9, 12). This is in agreement with sex-biases seen in other lung diseases such as influenza (13) and COVID-19 (14) and might relate to fundamental sex-differences that are minted in utero.

It is true that boys are more likely to develop asthma in childhood than girls. However, I feel that this fact is an insufficient reason for this study to be analyzed separately for boys and girls. I suggest that this part be deleted.

This section has been removed in accordance with your suggestion.

2. line 70-71

Animal models indicate that environmental insults during pregnancy program male and female fetuses differently (15, 16), though it is unclear to what extent this applies to humans.

This is probably the reason why this study was conducted separately for males and females. It is known that males are more prone to epigenetic mutations caused by environmental factors than females. Didn't you assume that it would be easier to identify the environmental factors that lead to epigenetic variation if you analyzed only males?

Not only in animal models, but also in human studies, it has been found that males are more prone to epigenetic mutations caused by environmental factors than females. I would suggest adding the following references for example.

Within-pair differences of DNA methylation levels between monozygotic twins are different between male and female pairs

BMC Med Genomics. 2016; 9: 55.

https://pubmed.ncbi.nlm.nih.gov/27561550/

This was a very interesting read. The referenced article has been added. Please see line 67-69.

3. line 166-167

Following backward elimination, eight of 23 variables remained in the Poisson regression model (AIC = 2936.429; p < 10-9 ; Table 2).

Based on the hypothesis that men are more strongly affected by epigenetics, it would be better to use only men in the selection of risk factors. I would like you to discuss this with the statisticians. Furthermore, it is puzzling that smoking during pregnancy was not included in the eight risk factors selected. This problem will be solved if smoking is found to be associated with asthma by analysis of males only.

We have divided our response into two parts, corresponding to your two comments.

SMOKING AS A PREDICTOR:

It is true that, while smoking was included as a potential variable in the backward selected Poisson regression model for all patients (male & female), it did not remain among the final set of potential risk factors. Hence the effect of smoking was not analyzed in the sex-specific Poisson regression models. 

In our analysis we applied backward selection to limit the number of variables in the model to such a degree that we only conserved a highly parsimonious model. Backward selection examines models following very strict variable inclusion/exclusion ordering, and, as a result, many other equally sensible models can be ignored in the process. As you point out, one could speculate that smoking is more closely associated with asthma in males than in females and that its’ correlation strength in the general (male + female) regression model therefore might be diluted to such an extent that it is excluded from the model.

However, we already provide complementary analyses in the form of APES (APproximated Exhaustive Search) that suggest that this is not the case. During APES analysis, we performed repeated subsampling in attempt to stabilize the variable selection process, this allowed us to calculate the frequency for which a certain variable was included as a predictor in a model of a given size. APES was applied to the general model (male + female) but also the sex-specific models (male only and female only). APES analysis shows that smoking was infrequently selected as a predictor in all three models (male & female, male only, female only). This is shown in Supplementary Figures S8 – S10. Since the repeated bootstrap subsampling during the APES analysis did suggest smoking to be a poor predictor for asthma in our population, we have no reason to believe that smoking should have been included as a predictor for the number of asthma admissions in the backward eliminated Poisson regression models for our population.

SELECTION OF RISK FACTORS BASED ON MEN ONLY:

As mentioned above, we have during APES analysis selected risk factors based on only male, only female, and male and female combined. When comparing variables that were selected more than 60% of times in models of reasonable statistical quality (evaluated by the Akaike Information Criterion), the variables selected by APES for “only male” and “only female” models closely resemble the variables that achieved significance in the sex-specific Poisson regression models. 

For example, predictive variables that were selected frequently (>60%) by APES when analyzing males only include BMI, allergic status, threatened premature labor, breastfeeding, maternal diabetes. Except for maternal diabetes, all of these variables have been included in the Poisson regression model for males and we hence find the Poisson regression model to be fairly complete.

4. line 262-264

Mechanistically, it is possible that the intrauterine environment is permanently altered through pregnancies and that these alterations influence female foetuses to a greater extent than male foetuses.

This speculation contradicts the hypothesis that men are more prone to epigenetic mutations caused by environmental factors than women. I feel that it is unnecessary to discuss female risk factors in order to clarify the intent of this study.

This speculation has been removed.

Specific recommendations for revision minor:

1. line 248-250

Instead, it is imaginable that medications used to treat TPL could alter the intrauterine environment and thereby affect foetal lung development.

I propose to add the following paper.

Beta-2 receptor agonist exposure in the uterus associated with subsequent risk of childhood asthma

Pediatr Allergy Immunol. 2017; 28: 746-753.

https://pubmed.ncbi.nlm.nih.gov/28892561/

Thank you for this recommendation, we have added this paper and rephrased this sentence. Please see line 264 – 266.

Reviewer #2: Interesting subject and good study. The key message here are sex differences in asthma admission rate and prenatal risk factors.

Minor remarks:

Abstract - statement “condition affecting boys more often than girls” – is not accurate while it’s true only before puberty – see ref. 10, and quite different later in life when women have more asthma than men. I will be helpful to add on some information form literature why those difference are observed - e.g. lung development, total IgE level.

This inaccuracy has been clarified in the abstract, see line 18-19. We have also added a brief section commenting on attributable factors to this phenomenon (e.g. IgE levels and airway diameter) to the introduction, see line 61-64. 

I suggest to add an information about the asthma admission cases – what is the age, sex and how many admissions. As I believe it’s 17 years period of time, so we have both 2 y. and 17 y. children?

Since asthma is a different phenotype before 5y. and after – it’s important to differentiate according to age. So called “childhood asthma” is triggered by infections, which are more common earlier in life in second child, than in first, and some young children outgrow symptoms later in life.

We have now added an additional supplementary figure (Figure S1) showing the age distributions for children with no, one, two, and three or more asthma admissions. This figure shows that the majority of included children is relatively young as also indicated by Table 1. Expectedly, children that were admitted several times tended to be slightly older but still the majority was prepubertal (judging by age). Please see line 164 – 166.

Socioeconomic status is an important risk factor affecting asthma admission related mainly to regularity of treatment but also smoking and obesity. The lack of this data is a disadvantage of this study, however possibly doesn’t affect the sex differences which were observed.

We agree that socioeconomic status is an important risk factor, unfortunately the design of this study did not allow us access to such data though. As you point out, we judge that it is unlikely to affect the sex-differences, as we do not assume that there is an underlying sex-bias in socioeconomic status.

In table 2 the data regarding admission is missing? - 0-0? Or not clear.

Table 2 shows the Poisson regression outcomes. In these regression models’ number of admissions was used as the response variable. This was stated in line 129 – 131 (Methods). We have further clarified the nature of the regression model in line 174-175 (Results), which we believe will increase the understandability of this section.

Assuming “0 - 0” in your comment refers to Table 1, this table shows the number of asthma admissions per child in the format: median [1st quartile – 3rd quartile]. Showing that children in both groups were admitted 0 [0-0] indicates that a majority of children in both the included and excluded group never was admitted for asthma. 

Kind Regards

Ralph Nanan

---

## [Decision Letter · Decision Letter 1]

12 May 2021

Modelling Maternal and Perinatal Risk Factors to Predict Poorly Controlled Childhood Asthma

PONE-D-21-03952R1

Dear Dr. Nanan,

We’re pleased to inform you that your manuscript has been judged scientifically suitable for publication and will be formally accepted for publication once it meets all outstanding technical requirements.

Kind regards,

Kazumichi Fujioka

Academic Editor

PLOS ONE

Additional Editor Comments (optional):

Reviewers' comments:

Reviewer's Responses to Questions

**Comments to the Author**

1. If the authors have adequately addressed your comments raised in a previous round of review and you feel that this manuscript is now acceptable for publication, you may indicate that here to bypass the “Comments to the Author” section, enter your conflict of interest statement in the “Confidential to Editor” section, and submit your "Accept" recommendation.

Reviewer #1: All comments have been addressed

Reviewer #2: All comments have been addressed

2. Is the manuscript technically sound, and do the data support the conclusions?

Reviewer #1: Yes

Reviewer #2: Yes

3. Has the statistical analysis been performed appropriately and rigorously? 

Reviewer #1: Yes

Reviewer #2: Yes

4. Have the authors made all data underlying the findings in their manuscript fully available?

Reviewer #1: Yes

Reviewer #2: No

5. Is the manuscript presented in an intelligible fashion and written in standard English?

Reviewer #1: Yes

Reviewer #2: Yes

6. Review Comments to the Author

Reviewer #1: (No Response)

Reviewer #2: All my remarks were addressed and vague aspects clarified. The manuscript has been improved. I don't have any further comments.

7. PLOS authors have the option to publish the peer review history of their article (what does this mean?). If published, this will include your full peer review and any attached files.

Reviewer #1: **Yes: **Mitsuhiro Okamoto

Reviewer #2: No

---

## [Editor Report · Acceptance letter]

18 May 2021

PONE-D-21-03952R1 

Modelling maternal and perinatal risk factors to predict poorly controlled childhood asthma 

Dear Dr. Nanan:

I'm pleased to inform you that your manuscript has been deemed suitable for publication in PLOS ONE. Congratulations! Your manuscript is now with our production department. 

Kind regards, 

on behalf of

Dr. Kazumichi Fujioka 

Academic Editor

PLOS ONE